# Preoperative Osteopenia Is Associated with Significantly Shorter Survival in Patients with Perihilar Cholangiocarcinoma

**DOI:** 10.3390/cancers14092213

**Published:** 2022-04-28

**Authors:** Jun Watanabe, Atsushi Miki, Yasunaru Sakuma, Kentaro Shimodaira, Yuichi Aoki, Yoshiyuki Meguro, Kazue Morishima, Kazuhiro Endo, Hideki Sasanuma, Alan Kawarai Lefor, Takumi Teratani, Noriyoshi Fukushima, Joji Kitayama, Naohiro Sata

**Affiliations:** 1Department of Surgery, Division of Gastroenterological, General and Transplant Surgery, Jichi Medical University, Shimotsuke 329-0498, Japan; m06105jw@jichi.ac.jp (J.W.); naruchan@jichi.ac.jp (Y.S.); ken_shimodaira@jichi.ac.jp (K.S.); uand1a@jichi.ac.jp (Y.A.); y-meguro@ho-saisei.jp (Y.M.); morishim@jichi.ac.jp (K.M.); kendo@jichi.ac.jp (K.E.); h-ssnm@jichi.ac.jp (H.S.); alefor@jichi.ac.jp (A.K.L.); teratani@jichi.ac.jp (T.T.); kitayama@jichi.ac.jp (J.K.); sata2018@jichi.ac.jp (N.S.); 2Department of Pathology, Jichi Medical University, Shimotsuke 329-0498, Japan; nfukushima@jichi.ac.jp

**Keywords:** biomarker, bone mineral density, hilar cholangiocarcinoma, osteopenia, prognostic, sarcopenia, survival

## Abstract

**Simple Summary:**

Perihilar cholangiocarcinoma is an infrequent and advanced hepatobiliary neoplasm with a generally poor prognosis. Surgical treatment is the only curative therapy that offers the promise of long-term survival. The identification of risk factors and appropriate treatment strategies are essential to improve long-term survival. Osteopenia is defined as low bone mineral density and has been shown to be associated with outcomes of patients with various cancers. The association between osteopenia and perihilar cholangiocarcinoma is unknown. This is the first report to show that osteopenia is associated with shorter survival in patients with perihilar cholangiocarcinoma. Preoperative osteopenia may be a useful tool for predicting prognosis in patients with perihilar cholangiocarcinoma.

**Abstract:**

Background: Osteopenia is defined as low bone mineral density (BMD) and has been shown to be associated with outcomes of patients with various cancers. The association between osteopenia and perihilar cholangiocarcinoma is unknown. The aim of this study was to evaluate osteopenia as a prognostic factor in patients with perihilar cholangiocarcinoma. Methods: A total of 58 patients who underwent surgery for perihilar cholangiocarcinoma were retrospectively analyzed. The BMD at the 11th thoracic vertebra was measured using computed tomography scan within one month of surgery. Patients with a BMD < 160 HU were considered to have osteopenia and b BMD ≥ 160 did not have osteopenia. The log-rank test was performed for survival using the Kaplan–Meier method. After adjusting for confounding factors, overall survival was assessed by Cox′s proportional-hazards model. Results: The osteopenia group had 27 (47%) more females than the non-osteopenia group (*p* = 0.036). Median survival in the osteopenia group was 37 months and in the non-osteopenia group was 61 months (*p* = 0.034). In multivariable analysis, osteopenia was a significant independent risk factor associated with overall survival in patients with perihilar cholangiocarcinoma (hazard ratio 3.54, 95% confidence interval 1.09–11.54, *p* = 0.036), along with primary tumor stage. Conclusions: Osteopenia is associated with significantly shorter survival in patients with perihilar cholangiocarcinoma.

## 1. Introduction

Perihilar cholangiocarcinoma is a relatively infrequent neoplasm typically involving the confluence of the hepatic ducts [1]. Because these tumors are located at the confluence of the right and left bile ducts and are closely related to the portal vain, hepatic artery, they tend to invade major vascular structures, and liver parenchyma, resulting in a low resectability rate of 34–47% [2]. Surgical resection is the only potentially curative treatment for patients with perihilar cholangiocarcinoma, resulting in a median overall survival (OS) of about 35–40 months, which is far from satisfactory [3]. Extensive resections are associated with significant postoperative morbidity and mortality of 5–15% in Western centers [4,5]. Therefore, it is necessary to carefully consider the indications for surgery.

A number of studies have confirmed that malnutrition dramatically increases the incidence of postoperative complications in patients with cancer who undergo resection, negatively affects the efficacy of anticancer treatments, length of hospital stay and quality of life of patients, and also accelerates tumor progression, leading to poor survival [6]. Approximately 20–50% of surgical patients at the time of admission are diagnosed with malnutrition [7,8,9], causing muscle weakness, delayed wound healing, and immune system dysfunction [6,10]. On the basis of these findings, in recent years, several nutrition-based scores have been identified as possible prognostic markers for various malignancies. The neutrophil–lymphocyte ratio (NLR), controlling nutritional status score, prognostic nutritional index (PNI), sarcopenia, and osteopenia have been shown to predict prognosis for patients with various cancers in addition to tumor markers [6,11,12].

Among these nutritional markers, osteopenia, a condition in which bone mineral density (BMD) is lower than normal but not as low as in osteoporosis, has recently received attention in association with hepatobiliary cancer [13,14,15]. A study showed for the first time that osteopenia, indicated by low BMD acquired from computed tomography (CT) scan data, is associated with mortality after liver transplantation for hepatocellular carcinoma [13]. In addition, an inverse relationship between osteopenia and survival after resection of extrahepatic biliary malignancies has been reported [16]. However, the relationship between osteopenia and outcomes in patients with perihilar cholangiocarcinoma has not been reported. The aim of present study was to clarify the impact of osteopenia on OS in patients with perihilar cholangiocarcinoma. The study hypothesis is that osteopenia is an independent risk factor for decreased OS in patients with perihilar cholangiocarcinoma.

## 2. Materials and Methods

### 2.1. Patients

Patients who underwent resection of perihilar cholangiocarcinoma between August 2007 and August 2021 at Jichi Medical University were included in this study. Patients with the presence of other active primary malignant diseases were excluded (*N* = 1). The protocol for this study was approved by the Institutional Review Board and conforms to the provisions of the Declaration of Helsinki (A20-110). The standard treatment was biliary confluence resection combined with major hepatectomy and lymphadenectomy. For some patients, S1 was prescribed as adjuvant chemotherapy. Recurrence was confirmed as locally or distant metastatic tumors by CT scan findings. For patients with recurrence, treatment was selected based on the performance status of the patient. An R0 resection was defined as complete resection of the grossly visible tumor and microscopically negative margins, while an R1 resection was defined as microscopically positive margins and an R2 resection was defined as incomplete resection of the grossly visible tumor. The lymph node staging of patients with perihilar cholangiocarcinoma followed the 7th edition of the American Joint Committee on Cancer (AJCC) staging system [17].

### 2.2. Detection of Osteopenia

BMD was evaluated by CT scan obtained within 1 month preoperatively. Non-contrast CT scan images at the 11th thoracic (T11) vertebral level were used. BMD was measured in the trabecular bone by calculating the average pixel density within a circle in the mid-vertebral core as previously described (Figure 1) [18]. The cut-off value was set as 160 Hounsfield units (HU) because the threshold of 160 HU in CT scan showed 90% sensitivity for osteoporosis using dual-energy X-ray absorptiometry as the reference standard in previous studies [13,18,19,20]. Patients with BMD < 160 HU were considered to have osteopenia and BMD ≥ 160 were classified as no osteopenia.

### 2.3. Definition of Sarcopenia

A low psoas muscle mass index (PMI), which indicates low skeletal muscle quantity, was measured using preoperative CT scan images at the level of the third lumbar (L3) vertebra. The PMI was calculated by dividing the cross-sectional area of the psoas muscle by the square of the height (cm^2^/m^2^) [16,18]. Gender-specific PMI cut-off values were previously established using data from healthy young Asian adults [21]. The PMI cut-off value used was 6.36 in males and 3.92 in females [21].

### 2.4. Measurement of NLR and PNI

NLR was calculated as the neutrophil count divided by the lymphocyte count using preoperative blood test results. The NLR cut-off value was 1.43 (area under the curve = 0.53) using a receiver operating characteristic analysis of overall survival [6,22]. PNI was calculated using: 10 × serum albumin value (g/dL) + 0.005 × lymphocytes (/mm^3^), with PNI less than 40 indicating impaired nutrition [23].

### 2.5. Statistical Analysis

Continuous variables were expressed as mean values ± standard deviation. All categorical data were analyzed using Pearson′s chi-square test or the Mann–Whitney U-test. Normally distributed values were analyzed by Student′s t-test. The OS curves in patients with or without osteopenia were constructed using the Kaplan–Meier method. A log-rank test was performed for survival using the Kaplan–Meier method. To compare OS by osteopenia, we evaluated the descriptive statistics and univariable analysis to isolate potential confounders. Continuous variables were analyzed as categorical variables, above and below a given cutoff. The cutoff values for serum cancer embryonic antigen and carbohydrate antigen 19-9 were 5 and 37, respectively, based on a previous report [24]. The hazard ratio (HR) and 95% confidence interval (CI) for OS were calculated using a Cox′s proportional hazard model. For multivariable model selection, a backward stepwise method using the Akaike information criterion (AIC) was employed to improve model fit [25]. All data were analyzed by SPSS version 25.0 (IBM Corp., Armonk, NY, USA) and R software packages (version 4.1.2; R Development Core Team). The level of statistical significance was set as *p* < 0.05.

## 3. Results

### 3.1. Patient Characteristics

There was no postoperative mortality during the study. Of 58 patients, a total of 33 patients survived. Median follow-up was 23 (range, 3–96) months. The median BMD values were 160 (interquartile range, 137–188) HU for men and women.

Table 1 shows the clinicopathological factors of patients with and without osteopenia. Of 58 patients, 27 had a low BMD and 31 had a normal BMD. The low BMD group had more females than the normal BMD group (*p* = 0.036). The two groups did not significantly differ for other characteristics, surgical factors, or tumor pathology.

### 3.2. Survival

The median survival of patients in the osteopenia group was 37 months and in the non-osteopenia group was 61 months (*p* = 0.034) (Figure 2).

In univariable analysis, primary tumor stage (HR 2.43, 95% CI 1.07–5.50, *p* = 0.033) and a low BMD (HR 2.42, 95% CI 1.04–5.60, *p* = 0.040) were associated with OS in patients with perihilar cholangiocarcinoma. The choice of the AIC-based statistical model revealed that osteopenia, gender, and primary tumor stage were the explanatory variables for the optimal model. In multivariable analysis, a low BMD was a statistically significant independent risk factor associated with shorter OS (HR 3.54, 95% CI 1.09–11.54, *p* = 0.036), along with primary tumor stage (HR 3.54, 95% CI 1.09–11.54, *p* = 0.029) (Table 2).

## 4. Discussion

These results show that osteopenia is an independent risk factor for shorter OS in patients with perihilar cholangiocarcinoma. Many prognostic factors for perihilar cholangiocarcinoma have been investigated, but, to our knowledge, this is the first report that osteopenia is associated with a negative impact on long-term survival in patients with perihilar cholangiocarcinoma. Since BMD can be measured preoperatively, it is useful because the prognosis can be predicted based on preoperative information. It may be useful as a treatment guideline, to help with decisions regarding the need for preoperative chemotherapy or postoperative adjuvant chemotherapy for patients with osteopenia, although this will require further evaluation.

The poor prognosis of patients with osteopenia with perihilar cholangiocarcinoma is controversial in relation to sarcopenia. In an earlier study, osteopenia was an independent risk factor for poor prognosis preceding sarcopenia [13]. In a previous systematic review, osteopenia was associated with all cancers except pancreatic cancer [26]. The present study of patients with perihilar cholangiocarcinoma shows that osteopenia may precede sarcopenia as a poor prognostic factor. Further studies are needed to elaborate on the prognostic impact of both osteopenia and sarcopenia.

The etiology of the progression of sarcopenia or osteopenia in patients with advanced malignancies is complicated. The mechanism of the impact of osteopenia on shorter survival is not understood, but one possible reason is that cachexia may stimulate osteoclasts and cause osteopenia [27,28,29,30,31,32]. Qui et al. have shown the NF-κB may be associated with the cause of sarcopenia [27]. A receptor activator of NF-κB ligand (RANKL) activates, while osteoprotegerin inhibits, osteoclastogenesis [28]. RANK is also expressed in skeletal muscle and activation of the NF-κB pathway mainly inhibits myogenic differentiation, which leads to skeletal muscle dysfunction and loss [29,30]. NF-κB promotes migration and invasion by upregulating Snail and the consequent repression of E-cadherin in cholangiocarcinoma cells [31]. NF-κB is reportedly a key molecule in the progression of biliary malignancies [32].

Measurement of BMD is a proxy for cumulative exposure to multiple factors, including vitamin D and estrogen. Obstructive jaundice can result from impaired absorption of lipid soluble vitamins. Vitamin D malabsorption leads to lower BMD. The influence of vitamin D on distant metastases was recently reported in pancreas cancer [33]. Several reports have demonstrated that sufficient vitamin D levels can inhibit tumor progression by maintaining the surroundings of pancreas ductal carcinoma in murine models [14]. Since changes in BMD were significantly associated with vitamin D levels, this may suggest a role for vitamin D in tumor progression [14]. In addition, estrogen was associated with a decreased risk for the development of cholangiocarcinoma [34]. This suggests that vitamin D and estrogen play important biological roles in the progression of perihilar cholangiocarcinoma. Further investigation of the relationship between vitamin D and estrogen profiles and osteoporosis in patients with perihilar cholangiocarcinoma is warranted.

The results of the present study suggest that perihilar cholangiocarcinoma surgery in physically active patients (which limits the development of osteopenia) yields improved outcomes. The treatment of osteopenia is not yet established for patients with cancer. Surgical and oncological outcomes may be improved by administering rehabilitation and nutrition therapy, including the use of symbiotics, micronutrients, branched-chain amino acids, vitamin D, calcium, and anti-osteoporotic drugs including bisphosphates, teriparatide, and denosumab, before surgery [18]. A bisphosphonate, zoledronate, has shown to prevent fractures in women with osteopenia [35]. Whether it has a similar effect in patients with resectable perihilar cholangiocarcinoma is unknown, but further studies should be conducted to identify effective supportive therapy.

Perihilar cholangiocarcinoma can be treated surgically, or, if surgical resection is not possible, by nonsurgical methods such as chemotherapy or palliative stenting. Unfortunately, in previous studies, only about one quarter of all patients underwent resection. This small number precludes validation of findings regarding sarcopenia and osteopenia assessed by CT scan and survival in patients with perihilar cholangiocarcinoma who undergo surgery [13,14,15,16,26]. This study is valuable as the first to identify osteopenia as a preoperative prognostic factor for patients with perihilar cholangiocarcinoma, although the sample size is somewhat small. Future larger studies are needed to evaluate the impact of osteopenia on survival.

Gender differences are important to consider because women generally have lower bone density compared to men [36]. In a systematic review of low BMD in patients with breast cancer, estrogen levels are related to some extent to BMD [37,38]. In this study, women were more often in the low BMD group than in the normal BMD group. Since the multivariable analysis adjusted for gender differences, the present review directly assessed the impact of low BMD on prognosis.

The presence of lymph node metastases is an important independent prognostic factor for OS [39]. However, some studies have identified the presence of lymph node metastases adjacent to the celiac artery only as an independent prognostic factor for OS [40]. Other studies have not demonstrated the prognostic value of regional lymph node metastases [41,42,43,44]. In the present study, the presence of regional lymph node metastases was not identified as an independent prognostic factor. In the 8th edition of *The AJCC*
*Cancer Staging Manual*, lymph node staging in patients with perihilar cholangiocarcinoma was changed and the N1 and N2 stage categories were modified based on the number of positive lymph nodes (N1: metastasis to 1–3 regional lymph nodes; N2: metastasis to 4 or more regional lymph nodes) [45]. Although we classified lymph node metastases according to the 7th edition of *The AJCC Cancer Staging Manual*, the 8th edition showed no significant benefit compared to the 7th edition in predicting prognosis of patients undergoing hepatic resection for perihilar cholangiocarcinoma in an earlier study [46]. Additionally, in the present study, only one patient was Stage N2 disease and the results showed that lymph node metastases were not associated with OS (HR 1.17, 95% CI 0.77–1.73, *p* = 0.436).

In a European center, the 90-day postoperative mortality was reported to be greater than 10% [47,48], while in Japan, the 90-day postoperative mortality was only 4% [49]. Another institution in Japan reported a mortality rate of 0 out of 40 consecutive patients in 2002 [50]. There are significant differences in patient characteristics, treatment strategies, perioperative outcomes, and survival between Eastern and Western countries [51]. Therefore, further studies in Western countries are needed to confirm these results.

In this study, there are several limitations. First, this is a retrospective study conducted at a single center with a relatively small sample size. However, earlier studies with small numbers of cases have also shown the prognostic values of osteopenia [18]. In addition, preoperative CT scan and laboratory blood tests were routinely performed within 1 month before surgery which limits the risk of observation bias. The results presented here should be prospectively validated in a larger population at multiple centers. Second, the cutoff value for diagnosing osteopenia determined is based on reports in the literature; however, further studies are necessary to verify whether the values are appropriate. Third, although we assessed the impact of each factor on osteopenia using backward stepwise multivariable logistic regression analysis with the AIC, dependencies among these variables may not be completely eliminated. Further studies with a large cohort are needed to fully adjust for confounding factors and validate the results.

## 5. Conclusions

This is the first report to show that osteopenia is associated with shorter survival in patients with perihilar cholangiocarcinoma. Preoperative measurement of BMD may be a useful tool for predicting prognosis in patients with perihilar cholangiocarcinoma.

## Figures and Tables

**Figure 1 cancers-14-02213-f001:**
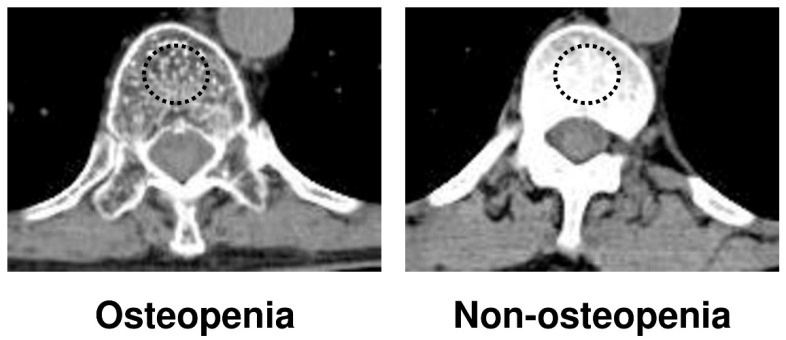
Measurement of bone mineral density on trabecular bone with calculation of the average pixel density within a circle in the mid-vertebral core at the 11th thoracic vertebral (T11) level.

**Figure 2 cancers-14-02213-f002:**
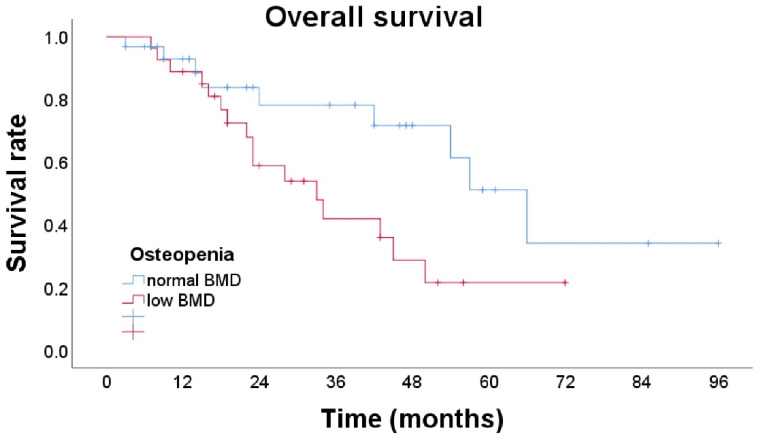
Overall survival of patients with and without osteopenia. The median survival of patients with osteopenia was 37 months and those without osteopenia had a significantly longer median survival of 61 months.

**Table 1 cancers-14-02213-t001:** Clinicopathological factors of patients with and without osteopenia.

Variables	Osteopenia *N* = 27	Non-Osteopenia *N* = 31	*p*-Value
Variables			
Age (y), mean ± SD	70.3 ± 7.2	69.0 ± 9.4	0.544
Gender (male/female), *N*	16/11	26/5	0.036 *
BMI (kg m^−2^), mean ± SD	22.3 ± 3.4	22.2 ± 3.2	0.991
ASA-PS ≥ 3, *N*	4/23	6/25	0.653
PMI (cm^2^/m^2^), mean ± SD	3.67 ± 1.4	3.99 ± 1.6	0.423
White blood cell (/mm^3^), mean ± SD	5767 ± 2560	5674 ± 1378	0.868
Serum albumin (g/dL), mean ± SD	3.69 ± 0.6	3.66 ± 0.6	0.829
PNI, mean ± SD	42.4 ± 6.9	44.1 ± 7.1	0.355
NLR, mean ± SD	2.67 ± 1.7	2.61 ± 2.4	0.921
CEA (mg/dL), mean ± SD	5.87 ± 14.0	2.87 ± 4.6	0.295
CA19-9 (IU/mL), mean ± SD	228 ± 505	447 ± 1141	0.346
Adjuvant chemotherapy (yes/no), *N*	5/22	4/27	0.556
Clavien–Dindo classification ≥ 3, *N*	15/11	17/14	0.829
Operative factor			
Operation time (min), mean ± SD	543 ± 111	569 ± 135	0.440
Intraoperative bleeding (mL), mean ± SD	1604 ± 1210	1337 ± 912	0.354
Procedure (right/left hepatectomy/others)	11/14/2	17/14/0	0.220
Pathological factor			
Maximum tumor size (mm), mean ± SD	36.1 ± 25.3	38.5 ± 25.9	0.751
Primary tumor stage (T3, T4/Tis, T1, T2), *N*	10/17	9/22	0.517
Lymph node metastases (yes/no)	10/17	8/23	0.504
R0 resection (yes/no)	13/14	14/17	0.820

ASA-PS, American society of anesthesiologists–physical status; BMI, body mass index; IMAC, intramuscular adipose tissue content; PMI, psoas muscle mass index; PNI, prognostic nutritional index; NLR, neutrophil–lymphocyte ratio; SD, standard deviation; * indicates *p* < 0.05; y, year.

**Table 2 cancers-14-02213-t002:** Univariate and multivariable analysis of factors associated with overall survival in patients with perihilar cholangiocarcinoma.

Variables	Univariable Analysis		Multivariable Analysis	
	HR (95%CI)	*p* Value	HR	*p* Value
Age (≥65)	0.92 (0.40–2.11)	0.846		
Gender (M/F)	1.05 (0.42–2.65)	0.923	2.85 (0.95–8.55)	0.062
BMI (≥25)	1.71 (0.70–4.17)	0.242		
ASA-PS (≥3)	1.87 (0.62–5.60)	0.265		
PMI (low)	1.96 (0.26–14.6)	0.511		
PNI (<40)	1.15 (0.48–2.79)	0.755		
NLR (>3.37)	1.76 (0.40–7.78)	0.455		
CEA (>5)	2.36 (0.92–6.06)	0.074		
CA19-9 (>37)	1.58 (0.67–3.70)	0.295		
Adjuvant chemotherapy (yes)	1.90 (0.64–5.66)	0.250		
Osteopenia (BMD < 160)	2.42 (1.04–5.60)	0.040 *	2.57 (1.06–6.28)	0.038 *
Operative factor				
Bleeding (≥1000 mL)	1.82 (0.75–4.41)	0.183		
Operation time (≥500 min)	1.36 (0.54–3.42)	0.508		
R0 resection (no)	1.45 (0.65–3.22)	0.363		
CD classification (≥3)	1.39 (0.57–3.38)	0.465		
Pathological factor				
Tumor size (≥50 mm)	0.44 (0.10–1.89)	0.269		
Primary tumor stage T3–4	2.43 (1.07–5.50)	0.033 *	3.08 (1.21–7.90)	0.019 *
Lymph node metastases (yes)	1.25 (0.55–2.87)	0.596		

ASA-PS, American society of anesthesiologists–physical status; BMI, body mass index; CD, Clavien–Dindo; CI, confidence interval; PNI, prognostic nutritional index; NLR, neutrophil–lymphocyte ratio; BMD, bone marrow density; M, male; F, female; HR, hazard ratio; * indicates *p* < 0.05.

## Data Availability

Our database contains highly confidential data which may provide insight into clinical and personal information about our patients and lead to identification of these patients. Therefore, according to organizational restrictions and regulations, these data cannot be made publicly available. However, the datasets used and/or analyzed during the current study are available from the corresponding author on reasonable request.

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
