# Peer review of "Preoperative Osteopenia Is Associated with Significantly Shorter Survival in Patients with Perihilar Cholangiocarcinoma"

_cancers, 2022, doi:10.3390/cancers14092213_

Round 1
Reviewer 1 Report
Thank you for your interesting manuscript addressing osteopenia as an independent preoperative predictor of survival in perihilar cholangiocarcinoma. It is an important topic where further research is utterly needed. I think your research should be base for extended research, i.e. multi-center and prospective studies.
Prior to publication of the manuscript at hand please address the following points:
- Please elaborate on the selection criteria of the parameters used for the multivariate models (assuming these are the parameters significant in univariate analysis and those different in the Osteopenia and Non-Osteopenia group, but that should be explained in the manuscript)
- Minor: Description of Table 2 (line 151/152) is in the wrong position
- Minor: line 175 and 179 it should be RANKL instead of RANKEL
Reviewer 2 Report
The paper is nicely written in good Enlish language. The suggestion that osteopenia is related to OS is understandable. Osteopenia may be a surrogate marker of age and sex, and perhaps an indicator of general performance status. Or –the hypothesis of this paper- an independent factor by itself.
L81 2007-2021 = 14yrs, n=58 > 4 per year is a very low number
127: not a single post-operative 90day mortality in 14 years? Generally speaking one would expect a rate of 5% at least, probably realistic estimates are around 15% for western series, perhaps Asian lower. But zero is really unique.
table 1: you show a under 100% resectability rate, but only resected patients were included in this study…please explain
table 2, It looks as if most of the variables are categorical? So you tested above and under the given cutoff? Please explain why you did not enter these variables as continuous variables.
- What are the results if you test them as continuous variables?
table 2 in the univariate analysis gender is not significant at all, p=0.9. How does it end up in the multivariate analysis? I assume you did a stepwise modelling? At which p did you consider a variable from the univariate to be entered in the multivariate analysis? (0.10 is a common cut-off)
Did you test osteopenia to be independent from for instance age and gender? I think there is a high risk of confounding bias in this analysis. How does the Kaplan Meijer look for male vs female?
220 you entered a univariate non-significant factor (sex) in the multivariate model and then it became significant. Please consult a statistician to rule out dependency between these variables. Especially in a small group (n=58) unexpected effects occur in multivariate models.
How are the parameters in the univariate table compared between males and females?
It is very surprising that nodal status does not affect survival, this is in contrast with almost all previous data. Did you count the number of positive nodes? Did you specify N1 or N2? AJCC 7th or 8th edition? How does this relate to OS?
You define resectability as R1 or R2 in table 2. You probably mean resection margin? How was this defined?
Where is the R0 status in this analysis?
Where is the WHO or ASA status in this analysis? This should be known for all patients. This could be the surrogate marker for osteopenia and vice versa!!
173: you did not study these pathways, all suggestions are not backed by your data. Breast cancer which has a tendency to metastasize to bone is a different disease, In pCCA and especially this group with a high recurrence free (non metastasizing!) rate is (or better may be) incomparable.
204: this is in extensive bone-metastasising disease…be careful to suggest an analogy or similar effect for resectable pCCA with (as you show) high cure rates.
